# Multiscale Decomposition Prediction of Propagation Loss for EM Waves in Marine Evaporation Duct Using Deep Learning

**Hanjie Ji [1,2], Bo Yin [1,*], Jinpeng Zhang [2], Yushi Zhang [2], Qingliang Li [2] and Chunzhi Hou [2]**

1    School of Computer Science and Technology, Ocean University of China, Qingdao 266100, China
2    National Key Laboratory of Electromagnetic Environment, China Research Institute of Radiowave Propagation, Qingdao 266107, China
*    Correspondence: ybfirst@ouc.edu.cn

**Abstract:** A tropospheric duct (TD) is an anomalous atmospheric refraction structure in marine environments that seriously interferes with the propagation path and range of electromagnetic (EM) waves, resulting in serious influence on the normal operation of radar. Since the propagation loss (PL) can reflect the propagation characteristics of EM waves inside the duct layer, it is important to obtain an accurate cognition of the PL of EM waves in marine TDs. However, the PL is strongly non−linear with propagation range due to the trapped propagation effect inside duct layer, which makes accurate prediction of PL more difficult. To resolve this problem, a novel multiscale decomposition prediction method (VMD−PSO−LSTM) based on the long short−term memory (LSTM) network, variational mode decomposition (VMD) method and particle swarm optimization (PSO) algorithm is proposed in this study. Firstly, VMD is used to decompose PL into several smooth subsequences with different frequency scales. Then, a LSTM−based model for each subsequence is built to predict the corresponding subsequence. In addition, PSO is used to optimize the hyperparameters of each LSTM prediction model. Finally, the predicted subsequences are reconstructed to obtain the final PL prediction results. The performance of the VMD−PSO−LSTM method is verified by combining the measured PL. The minimum *RMSE* and *MAE* indicators for the VMD−PSO−PSTM method are 0.368 and 0.276, respectively. The percentage improvement of prediction performance compared to other prediction methods can reach at most 72.46 and 77.61% in *RMSE* and *MAE*, respectively, showing that the VMD−PSO−LSTM method has the advantages of high accuracy and outperforms other comparison methods.

**Keywords:** tropospheric duct; propagation loss; deep learning; LSTM network; multiscale decomposition prediction; VMD method; PSO algorithm

## 1. Introduction

A tropospheric duct (TD) is an abnormal atmospheric refraction structure appearing at the marine–atmosphere boundary [1], with ultra−long horizontal scale characteristics and a typical weather background. The main reason for its appearance is due to the rapid decrease of water vapor in the vertical gradient, which results in the decrease of atmospheric refraction index with the increase of height, i.e., there is a negative gradient relationship [2]. When the negative gradient relationship satisfies certain conditions, the TD will cause the propagation path of electromagnetic (EM) waves caught in it to bend towards the marine surface. When the curvature of EM waves bent towards the marine surface is smaller than the curvature of the earth, EM waves will be trapped in a thin atmospheric layer of a certain thickness above the marine surface for propagation. This phenomenon is called the TD propagation of EM waves [3]. In TDs, the propagation loss (PL) of radar EM waves is usually weaker than in the normal atmospheric environment, appearing over−the−horizon propagation, which extends the detection range of radar. However, as the EM waves are trapped inside duct layer, the EM energy is less distributed

in the upper area of the duct layer, resulting in unexpected holes and lowering the target detection performance of radar [4–6]. The main effects of marine TDs on the radar system are shown in Figure 1.

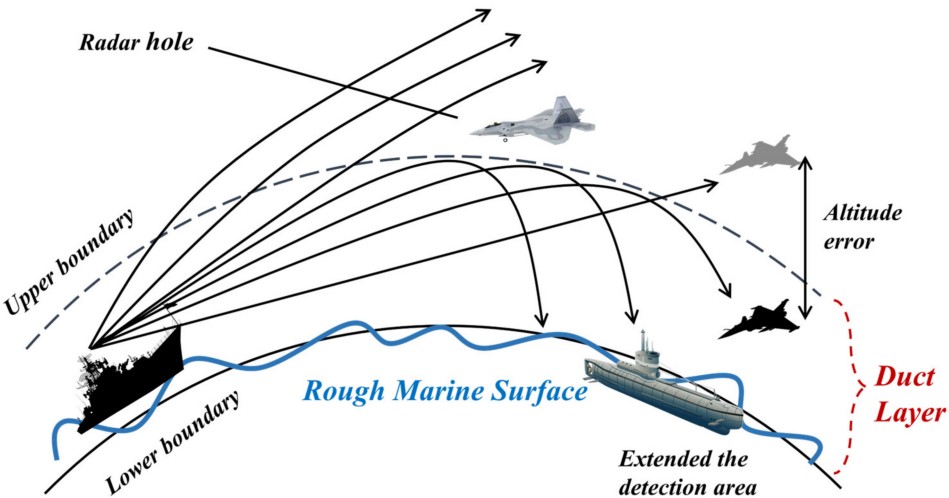

**Figure 1.** Main effects of a marine TD on radar systems.

Based on valid TD environmental information obtained, the precondition for an accurate assessment of the influence of a TD on the performance of a radar system is accurate cognition of the EM waves propagation characteristics [7]. Since the PL in a TD can reflect the trapped propagation effect of the duct layer on EM waves and the propagation characteristics of EM waves, the PL is of vital research and application value. However, in TDs, the propagation path and range of EM waves appear as significant anomalies relative to the standard atmospheric environment, making the accurate cognition of the propagation characteristics of EM waves inside the duct layer very difficult [8].

The main theoretical algorithms currently used to calculate EM wave PL in TDs include the ray tracing algorithm, duct mode algorithm, parabolic equation algorithm and multiple hybrid algorithms [9]. However, as the propagation range varies, the calculation time using theoretical algorithms increases rapidly and the calculation efficiency decreases, which does not satisfy the real−time requirements for engineering applications. In addition, when the TD phenomenon appears, shipboard radar can be used to conduct over−the−horizon detection of the marine surface and to obtain the PL on the EM waves' propagation path. However, when studying the actual propagation characteristics of EM waves, due to the limitations of measurement conditions and experimental equipment, it is usually not possible to obtain all the PL values along the entire propagation path. Generally, only the PL values on the propagation path closer to the radar can be measured [10].

Rapid development in deep learning offers new solutions for accurate prediction of PL. Recurrent neural network (RNN) has achieved outstanding results in sequence prediction since its proposal [11]. LSTM is an improved RNN with better generalization ability and fault tolerance, which can solve the problems of RNN's inability to achieve memorization and forgetting of long−term historical information [12]. However, building prediction model with better performance and achieving accurate prediction of PL is extremely difficult [13]. For a radar system, the trapped propagation effect in a TD results in EM waves with strongly non−linear and non−smooth characteristics for the following main reasons: (1) When EM waves are trapped inside the duct layer, they are influenced by the interference of the duct mode, which can result in jumping propagation due to mutual interference and elimination of multiple duct modes. Additionally, when the strength of the duct is strong, there are more duct modes inside duct layer, and the interference effects between them can be even more complicated. Moreover, as the wind velocity at marine surface varies, interference between duct modes also results in variations of the

position where PL will enhance and weaken [14]. (2) The weather's physical quantities such as temperature, humidity and pressure above a marine surface can have complicated variations in time and space due to various complicated or random weather processes. However, the refractivity in tropospheric atmosphere is a multivariate function relating to the physical quantities, thus the refractivity is usually non−uniform in time and space scales [15]. The different space distribution of the atmospheric reflectivity can seriously influence the propagation path of EM waves, making them have different space energy distributions, resulting in significantly different space propagation characteristics of EM waves compared to the standard atmosphere [16].

These reasons have a vast influence on the propagation characteristics of EM waves, making the PL seriously non−smooth and random. When the non−smooth characteristics are strong, the propagation characteristics cannot be accurately cognized, which will make the training of deep learning methods more difficult and the prediction accuracy lower. Therefore, for the measured PL in a TD, it is necessary to decompose the PL sequence into simple subsequences by the signal decomposition method before using deep learning method for prediction, to lower its non−smoothness and contribute to improving the prediction accuracy [17]. Typical signal decomposition methods include wavelet analysis and empirical mode decomposition (EMD). However, wavelet analysis relies on the choice of wavelet basis function and is a non−adaptive decomposition method, and EMD is prone to the problem of mode mixing, which can influence the decomposition results [18]. The newly proposed variational mode decomposition (VMD) method not only overcomes the shortcomings of wavelet analysis and EMD, but also effectively solves the problems such as strong non−smoothness of PL. VMD can decompose PL into multiple subsequences of different frequency scales and relative smoothness, i.e., intrinsic mode function (IMF) components [19]. Therefore, the LSTM network can be used to predict IMF components, and then reconstruct the corresponding prediction results, improving the prediction accuracy of the LSTM network. Currently, many heuristic optimization algorithms have been integrated into deep learning for prediction performance improvement. The particle swarm optimization (PSO) algorithm is an efficient optimization algorithm that finds the global optimal solution by continuously updating the velocity and position of particles, with the advantages of high accuracy and fast convergence [20]. PSO can effectively optimize the hyperparameters of the LSTM network such as the number of neurons and learning rate, thus improving the prediction performance of LSTM networks.

Based on the above study methods, this study proposes a novel hybrid prediction method (VMD−PSO−LSTM) by combining the VMD method, PSO algorithm and LSTM network based on measured PL to achieve accurate prediction in marine evaporation duct (ED). The VMD−PSO−LSTM method is constructed as follows: (1) using the VMD method, the measured PL is decomposed to obtain a limited number of smooth subsequences with different frequency scales; (2) for the decomposed subsequences (IMF components) of different scales, corresponding LSTM prediction models are built, respectively, using the PSO algorithm to optimize the hyperparameters of each LSTM prediction model, and using the optimized LSTM network to predict the subsequences; (3) the prediction results for each PL subsequences are reconstructed to obtain the final PL prediction results.

The remainder of this paper is constructed as follows: Section 2 introduces the marine over−the−horizon propagation experiment and measured PL. Section 3 introduces the modeling process and methods used in the VMD−PSO−LSTM method. The prediction results and analysis for multiple prediction methods are introduced in Section 4. Finally, the conclusions and future work are presented in Section 5.

## 2. EM Waves Over−the−Horizon Propagation Experiment in Marine ED

The marine ED is the TD type with the highest probability of appearance, which can reach up to 89% in specific marine areas, and often appears in the near marine surface atmosphere below 40 m of height and consists of a shallow trapped layer. In addition, the appearance probability of the ED and evaporation duct height (EDH) vary significantly

with geographical area, season and time of day. Typically, ED has a higher appearance probability and higher EDH during summer and daytime in low−latitude marine areas. Due to its weak trapping capability, the frequency of EM waves that can be trapped or significantly influenced is generally above 1 GHz [1–3].

In 2017, the China Research Institute of Radiowave Propagation conducted an over−the−horizon propagation experiment of EM waves in the South China Sea, which relied on a shipboard S−band radar installed on the "Qiongsha 3" ship [21]. The ship moved irregularly between Wenchang (19°33′ N, 110°49′ E) and Yongxing Island (16°84′ N, 112°33′ E). The marine area where the over−the−horizon propagation experiment was conducted is shown in Figure 2. The "Qiongsha 3" ship is 84 m long, 13.8 m wide and displaces about 2500 tons, as shown in Figure 3a. The experimental equipment installed on the deck is shown in Figure 3b. The radar was installed on the port side of the bow, about 12 m above the marine surface. Considering the height of the radar antenna, the total height of the radar is about 14 m above the marine surface and its signal is easily trapped by the marine ED. The depression angle of the S−band radar is set at 1°. The main parameters of the shipboard S−band radar are shown in Table 1.

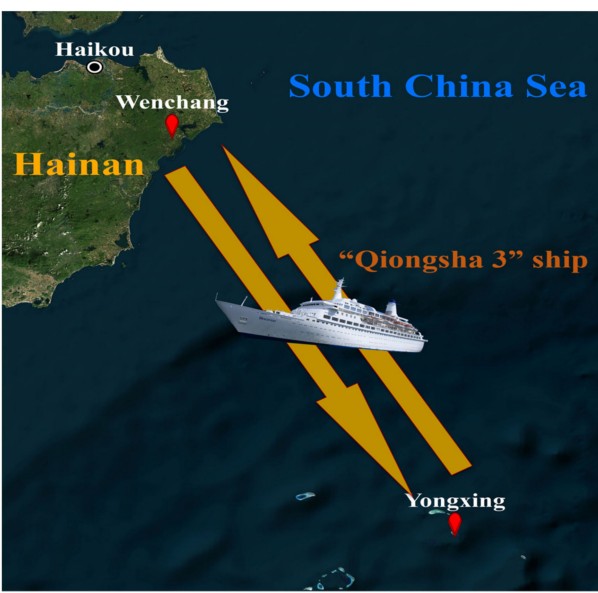

**Figure 2.** Marine area for the conduct of the over−the−horizon propagation experiment.

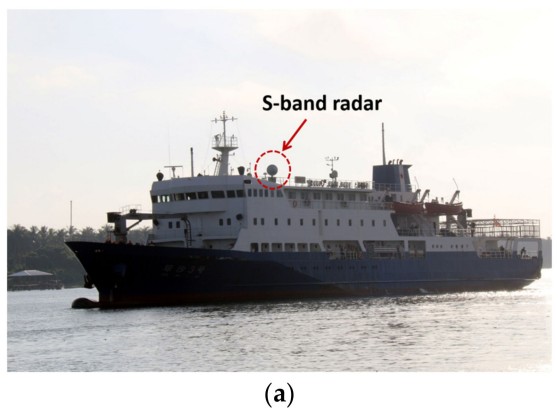

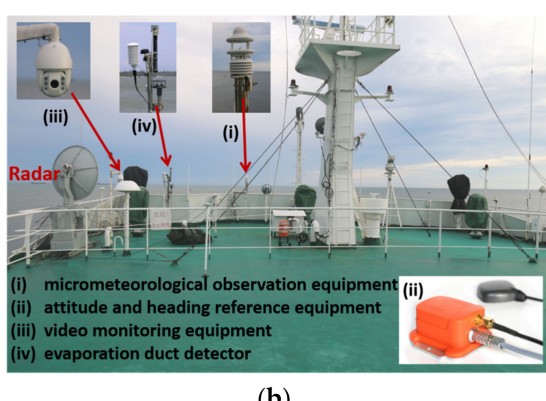

(**a**)                                                      (**b**)

**Figure 3.** Scene of the over−the−horizon propagation experiment. (**a**) "Qiongsha 3" ship and S−band radar; (**b**) equipment installed on deck.

**Table 1.** S−band radar system parameters in the over−the−horizon propagation experiment.

| Number | Parameter | Value |
|:---:|:---:|:---:|
| 1 | Frequency | 3.1 GHz |
| 2 | Transmitting Power | 41.8 dBm |
| 3 | Transmitting Antenna Gain | 28.0 dB |
| 4 | Pulse Repetition Frequency | 2.0 kHz |
| 5 | Bandwidth | 5.0 MHz |
| 6 | Pulse Width | 3.0 μs |
| 7 | Polarization | HH |

The experiment was conducted as follows: while the Qiongsha 3 ship was moving between Wenchang and Yongxing Island, the S−band radar continuously transmitted signals to the onshore receiver in Hainan. An EMI signal receiver was installed at the receiver for continuous monitoring and receiving the S−band over−the−horizon signals, and a low noise amplifier was installed at the receiver front to improve the signal−to−noise ratio. According to the radar equation [22], the one−way PL of EM waves is calculated as:

$$L = P_t + G_t + G_r + G_{LNA} - P_r - L_{r1} - L_{r2} \tag{1}$$

where $P_t$ and $P_r$ represent the radar transmitting power and receiver receiving power, respectively. $G_t$ and $G_r$ represent the transmitting antenna gain and the receiving antenna gain, respectively, $G_{LNA}$ represents the low noise amplifier gain. $L_{r1}$ and $L_{r2}$ represent the feedline loss at the transmitter and receiver, respectively. The above equipment parameters in the over−the−horizon propagation experiment are shown in Table 2.

**Table 2.** Equipment parameters in the over−the−horizon propagation experiment.

| Number | Parameter | Value |
|:---:|:---:|:---:|
| 1 | Receiving Antenna Gain | 16.0 dB |
| 2 | Low Noise Amplifier Gain | 22.3 dB |
| 3 | Transmitter Feedline Loss | 2.0 dB |
| 4 | Receiver Feedline Loss | 2.0 dB |

The over−the−horizon propagation experiment was conducted four times in total and three obvious sets of over−the−horizon propagation signals were collected. According to Equation (1), PL sequences of the three sets of over−the−horizon signals can be calculated by combining the equipment parameters, as shown in Figure 4.

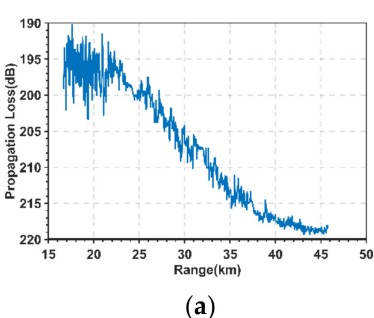

(a)

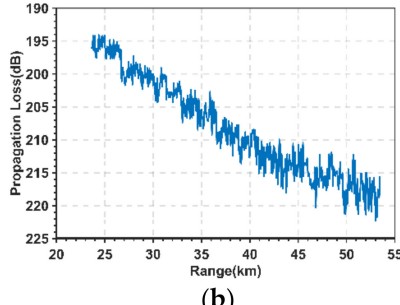

(b)

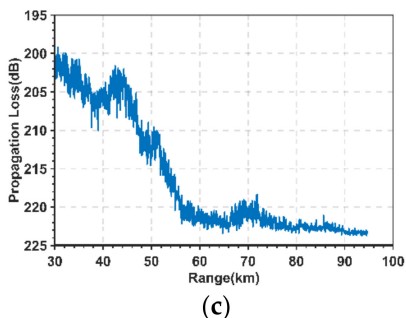

(c)

**Figure 4.** Three sets of PL sequences measured in the over−the−horizon propagation experiment: (**a**) first set; (**b**) second set; (**c**) third set.

From Figure 4, the variation trend of each PL at different ranges are various, because of the influence of the ED, the PL sequences have serious deviation and there are strong non−linear and non−smoothness characteristics, which cannot reflect the real propagation situations of EM waves. So, it is necessary to lower the non−smoothness of the PL

sequences, which is very essential to achieve the subsequent accurate prediction of PL sequences at long ranges [23].

Figure 5 shows the EDH data on the Wenchang–Yongxing Island path obtained while measuring the third set of PL [24]. Figure 5 shows that the EDHs at different measurement locations are varied over the corresponding time and space during the over−the−horizon experiment. Overall, the EDH is smaller when the ship is close to Wenchang, and when the ship moves away from Wenchang reaching a remote marine area, the EDH is larger and very undulating [25]. The different space distribution of EDH can seriously influence the propagation path of EM waves, resulting in PL with strong non−linear characteristics.

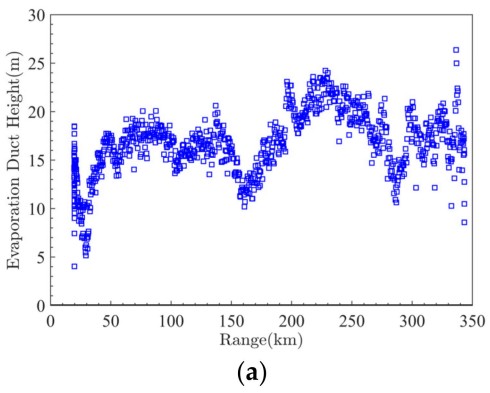
(**a**)

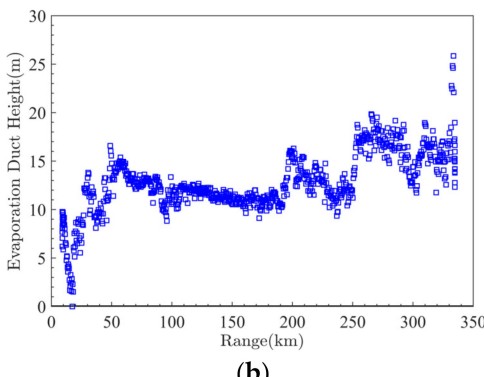
(**b**)

**Figure 5.** EDHs measured on Wenchang–Yongxing Island path: (**a**) Wenchang–Yongxing Island; (**b**) Yongxing Island–Wenchang.

### 3. Multiscale Decomposition Prediction of PL in ED

*3.1. General Framework of Multiscale Decomposition Prediction Model*

In this study, the VMD method, PSO algorithm and LSTM network, are used to build the VMD−PSO−LSTM model for PL prediction in marine EDs. The LSTM network is a special RNN that solves the problems of gradient explosion and short−term memory in RNN and can extract the data correlation inside the PL well [26]. Since the measured PL is a non−linear and highly complicated sequence, if the PL sequence is predicted directly, it is prone to generate large errors, which seriously influences the accurate cognition of EM waves' propagation characteristics. Therefore, the measured non−smooth PL is decomposed to obtain the IMF components with smoothness and a trend, thus improving the accuracy of LSTM network [27]. The VMD method is more effective than other signal decomposition methods in extracting the detailed characteristics of the PL and lowering the non−smoothness of the PL, which can contribute to building a model with higher prediction accuracy. In addition, this study uses the PSO algorithm, replacing the process of artificially setting the hyperparameters of the LSTM network based on experience with using the PSO algorithm to automatically search for the optimal hyperparameters [28]. Additionally, this study uses the LSTM network optimized by the PSO algorithm to predict the PL subsequence. Eventually all the PL subsequence prediction results are reconstructed as predicted values of PL in EDs. The general framework of the VMD−PSO−LSTM model is shown in Figure 6. The detailed PL prediction process of the VMD−PSO−LSTM model is as follows:

**(1)** Using VMD method to decompose PL. The measured PL is decomposed using the VMD method to obtain a limited number of IMF components. This study obtains the optimal number of components of the PL by calculating the central frequency, and the decomposition process is expressed as follows:

$$L(t) = \sum_{i=1}^{K} IMF_i(t) \tag{2}$$

where $L(t)$ and $IMF(t)$ represent the PL and IMF components, respectively, and $K$ represents the total number of IMF components.

**(2)**    Building IMF components prediction models. Initializing the PSO algorithm, randomly generating a population of particles and setting the parameters of the PSO algorithm. Subsequently, using the PSO algorithm to optimize the hyperparameters (number of neurons, number of iterations and the learning rate) of the LSTM network [29]. Using the optimized LSTM network to learn the history state information for each set of PL subsequences and predict the value of the corresponding PL subsequences. The prediction principle of the PL subsequence can be expressed as:

$$Y(t+1) = PLSTM[IMF(t), IMF(t-1)] \tag{3}$$

where $PLSTM(\cdot)$ represents the LSTM network optimized by PSO and $Y(t+1)$ represents the predicted value of the PL subsequence.

**(3)**    Reconstructing the IMF components' prediction result. The predicted values of each PL IMF component are reconstructed to obtain the final prediction result of PL in a marine ED. The reconstructed result can be expressed as:

$$L(t+1) = \sum_{i=1}^{K} Y_i(t+1) \tag{4}$$

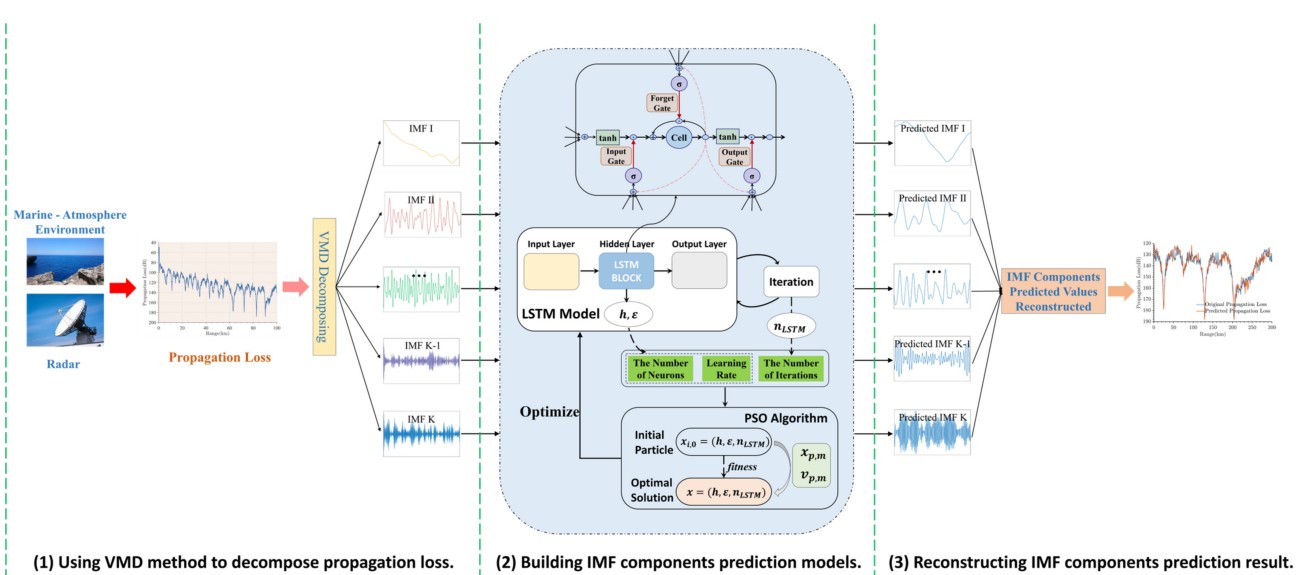

**Figure 6.** General framework of the VMD−PSO−LSTM model.

### 3.2. Using VMD Method to Decompose PL Sequence

The PL of EM waves in a marine ED is a type of non−smooth signal, so we use the VMD method to preprocess the PL to lower the complexities of the PL [30]. VMD is an adaptive and completely non−recursive signal analysis method based on the Wiener filter theory. VMD can decompose the PL into multiple subsequences called IMF components by setting the number of components for the PL, according to the characteristics of the PL in a real ED. Thus, VMD can achieve the effective decomposition of IMF components and has strong robustness to noise [31]. The VMD method can also effectively lower the non−smoothness of a complicated signal after the signal has been effectively reconstructed with valid information from different frequency bands. In the VMD method, each IMF component is defined as a non−smooth AM–FM signal, and the main decomposition process is as follows [32]:

**(1)** Conducting the Hibbert transform on each IMF component to obtain the unilateral frequency spectrum and the analytical signal of the IMF component:

$$\left[\delta(t) + \frac{j}{\pi t}\right] * u_k(t) \tag{5}$$

where "$*$" represents the convolution operation, $\delta(t)$ represents an impulse function, $j$ represents the imaginary part and $t$ represents time.

**(2)** Using the exponential operator $e^{-j\omega_k t}$ to correct the center frequency of the IMF component, and modulate the spectrum of IMF components to the corresponding baseband:

$$\left[\left(\delta(t) + \frac{j}{\pi t}\right) * u_k(t)\right] e^{-j\omega_k t} \tag{6}$$

**(3)** Solving the gradient of the demodulation signal, calculating the squared norm of the demodulation gradient and estimating the bandwidth of each IMF component, obtaining the following variational constraint model:

$$\begin{cases} \min\limits_{\{u_k\},\{\omega_k\}} \left\{ \sum_{k=1}^{K} \left\| \partial_t \left[ \left(\delta(t) + \frac{j}{\pi t}\right) * u_k(t) \right] e^{-j\omega_k t} \right\|_2^2 \right\} \\ s.t. \sum_{k=1}^{K} u_k(t) = L(t) \end{cases} \tag{7}$$

where $\{u_k\} = \{u_1, u_2, \dots, u_K\}$ represents the $K$ IMF components obtained from the decomposition of PL $L(t)$, $\{\omega_k\} = \{\omega_1, \omega_2, \dots, \omega_K\}$ represents the central frequency of each IMF component and $\partial_t$ represents the time derivative.

**(4)** For the above variational constraint model, the model can be converted to an unconstrained variational problem by introducing a penalty factor $\alpha$ and Lagrange operator $\lambda(t)$ to obtain its optimal solution, and the extended Lagrange expression is as follows:

$$\begin{aligned} L(\{u_k\}, \{\omega_k\}, \lambda) &= \alpha \sum_{k=1}^{K} \left\| \partial_t \left[ \left(\delta(t) + \frac{j}{\pi t}\right) * u_k(t) \right] e^{-j\omega_k t} \right\|_2^2 \\ &+ \left\| L(t) - \sum_{k=1}^{K} u_k(t) \right\|_2^2 + \left\langle \lambda(t), L(t) - \sum_{k=1}^{K} u_k(t) \right\rangle \end{aligned} \tag{8}$$

**(5)** Using the alternating direction multiplier method to continuously update each IMF component $u_k$ and central frequency $\omega_k$ to solve the "saddle point" (the optimal solution to the original variational constraint model) of the extended Lagrange expression, the updated IMF component and central frequency, respectively, are as follows:

$$u_k^{n+1}(\omega) = \frac{L(\omega) - \sum_{i<k} u_i^{n+1}(\omega) - \sum_{i>k} u_i^{n}(\omega) + \frac{\lambda^n(\omega)}{2}}{1 + 2\alpha(\omega - \omega_k^n)^2} \tag{9}$$

$$\omega_k^{n+1} = \frac{\int_0^\infty \omega \left| u_k^{n+1}(\omega) \right|^2 d\omega}{\int_0^\infty \left| u_k^{n+1}(\omega) \right|^2 d\omega} \tag{10}$$

$$\lambda^{n+1}(\omega) = \lambda^n(\omega) + \tau \left( L(\omega) - \sum_{k=1}^{K} u_k^{n+1}(\omega) \right) \tag{11}$$

where $\tau$ is the noise tolerance of the signal. Given the allowable error value $\zeta > 0$, iterating until the convergence condition is met or the maximum number of iterations is reached:

$$\frac{\sum_{k=1}^{K} \left\| u_k^{n+1} - u_k^n \right\|_2^2}{\sum_{k=1}^{K} \left\| u_k^n \right\|_2^2} < \zeta \tag{12}$$

In the VMD method, the decomposition results of the IMF components are mainly influenced by the number of components $K$ [33]. The waveshape and frequency of the decomposed IMF components will vary with the $K$. The more accurate the size of $K$ is set, the better the decomposed IMF components will be in describing the propagation characteristics of PL, and the more accurate the following prediction results obtained by using the LSTM network. Therefore, before decomposing the PL, $K$ needs to be set in advance; a too large value of $K$ will generate additional noise or result in mode mixing, too small will result in IMF components being under−decomposed and some important information will be filtered out. To make the decomposed IMF components describe as much information as possible about the characteristics of PL and to achieve more accurate prediction results [34], this study sets the best $K$ by calculating the central frequency distribution for different decomposition numbers, and the corresponding calculation process is shown in Figure 7.

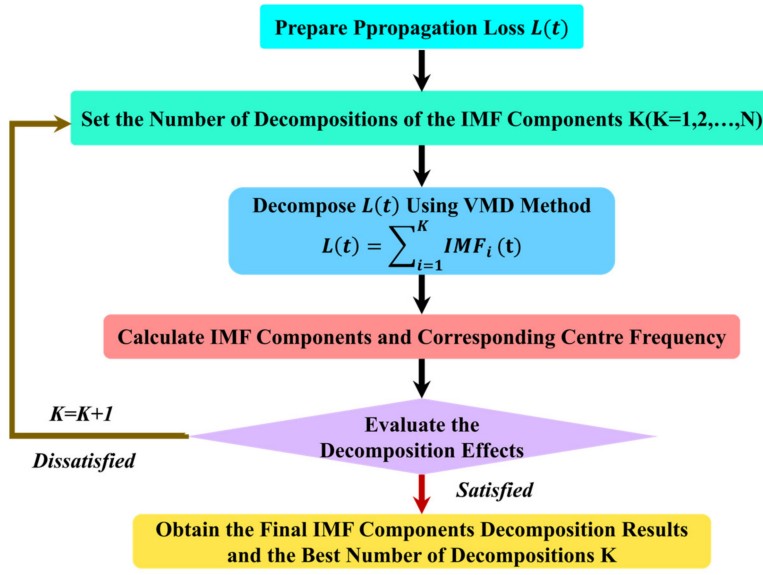

**Figure 7.** Process for calculating the central frequency of different decomposition numbers.

To balance the size of $K$ and the accuracy of the decomposition for the measured PL sequences, this study calculates and observes the central frequency distribution to determine the size of $K$ [35]. The central frequency of the IMF components obtained from the VMD decomposition are distributed from low frequency to high frequency. For a set of PL sequences, the number of components is calculated from small to large, to calculate the central frequency under different decomposition numbers. When the central frequency of the IMF component of the last layer remains relatively stable, it can be considered that the best size of $K$ is obtained at this time. Using the VMD method to decompose measured PL sequences under different $K$ values, the central frequency distribution of each IMF component (IMF1, IMF2, ... IMFK) is obtained as shown in Tables 3–5.

**Table 3.** Central frequency for the IMF components of the first set of PL.

| K | IMF1 | IMF2 | IMF3 | IMF4 | IMF5 | IMF6 | IMF7 |
|---|------|------|------|------|------|------|------|
| 2 | $7.84 \times 10^{-7}$ | 0.085722 | | | | | |
| 3 | $7.77 \times 10^{-7}$ | 0.080441 | 0.204793 | | | | |
| 4 | $7.75 \times 10^{-7}$ | 0.079771 | 0.200860 | 0.335962 | | | |
| 5 | $7.74 \times 10^{-7}$ | 0.079503 | 0.199620 | 0.324704 | 0.436563 | | |
| 6 | $7.62 \times 10^{-7}$ | 0.071275 | 0.134334 | 0.210364 | 0.326630 | 0.439310 | |
| 7 | $7.47 \times 10^{-7}$ | 0.060998 | 0.106955 | 0.171296 | 0.238401 | 0.329121 | 0.437847 |

**Table 4.** Central frequency for the IMF components of the second set of PL.

| K | IMF1 | IMF2 | IMF3 | IMF4 | IMF5 | IMF6 | IMF7 | IMF8 |
|---|------|------|------|------|------|------|------|------|
| 2 | $7.11 \times 10^{-7}$ | 0.160041 | | | | | | |
| 3 | $7.00 \times 10^{-7}$ | 0.124964 | 0.264836 | | | | | |
| 4 | $6.97 \times 10^{-7}$ | 0.120772 | 0.252542 | 0.376416 | | | | |
| 5 | $6.48 \times 10^{-7}$ | 0.066249 | 0.16108 | 0.267773 | 0.416254 | | | |
| 6 | $6.44 \times 10^{-7}$ | 0.064167 | 0.156436 | 0.255209 | 0.352884 | 0.450286 | | |
| 7 | $6.20 \times 10^{-7}$ | 0.052778 | 0.129169 | 0.211793 | 0.283487 | 0.366564 | 0.455107 | |
| 8 | $6.01 \times 10^{-7}$ | 0.045661 | 0.108541 | 0.168596 | 0.240634 | 0.308612 | 0.379114 | 0.458929 |

**Table 5.** Central frequency for the IMF components of the third set of PL.

| K | IMF1 | IMF2 | IMF3 | IMF4 | IMF5 | IMF6 | IMF7 | IMF8 | IMF9 |
|---|------|------|------|------|------|------|------|------|------|
| 3 | $3.63 \times 10^{-7}$ | 0.110601 | 0.226735 | | | | | | |
| 4 | $3.62 \times 10^{-7}$ | 0.105327 | 0.216277 | 0.395508 | | | | | |
| 5 | $3.55 \times 10^{-7}$ | 0.085214 | 0.165108 | 0.317864 | 0.435583 | | | | |
| 6 | $3.48 \times 10^{-7}$ | 0.070245 | 0.151811 | 0.238644 | 0.333217 | 0.441332 | | | |
| 7 | $3.38 \times 10^{-7}$ | 0.054487 | 0.113588 | 0.223596 | 0.302816 | 0.384877 | 0.461104 | | |
| 8 | $3.64 \times 10^{-7}$ | 0.115827 | 0.233349 | 0.175235 | 0.236412 | 0.311493 | 0.389808 | 0.463149 | |
| 9 | $3.36 \times 10^{-7}$ | 0.052281 | 0.107843 | 0.166575 | 0.223360 | 0.285226 | 0.342295 | 0.405546 | 0.469048 |

As can be seen from Table 3, for the IMF components of the first set of PL, the central frequency of the IMF components of the last layer starts to remain stable after *K* is greater than 4. Therefore, when *K* is equal to 5, the central frequency of adjacent IMF components is more spaced. The results of central frequency decomposition are relatively better, which can effectively avoid the phenomenon of mode mixing and can well explore the characteristic information inside PL [36]. Similarly, we can obtain the best number of components for the second and third sets of PL sequences. The best number of components for three sets of PL sequences is shown in Table 6.

**Table 6.** Best number of components for three sets of measured PL sequences.

| Propagation Loss | Best Number of Components |
|---|---|
| First Set | 5 |
| Second Set | 6 |
| Third Set | 7 |

The decomposed IMF components of three sets of PL sequences are shown in Figures 8–10. From Figures 8–10, after the VMD decomposition, the IMF components can be completely decomposed from small to large according to the frequency. The decomposed IMF components are all different, which does not result in over−decomposition, showing that the decomposed IMF components characterize as much information as possible about the measured PL. Analysis of the IMF components shows that the IMF1 reflects the variation trend of the measured PL sequence. IMF2 is highly regular and contains more periodic information. Other IMF components are less regular and contain more non−periodic information. It shows that through VMD decomposition, both the variation trend and local characteristics inside the PL are explored, thus contributing to the prediction model to better learn the characteristics inside the PL and reduce the difficulty of prediction. Moreover, except for IMF1, the IMF components are stable, and the values are all relatively evenly distributed on both sides of 0. Thus, the purpose of filtering out noise is also achieved by using VMD, reducing the influence of noise on prediction accuracy.

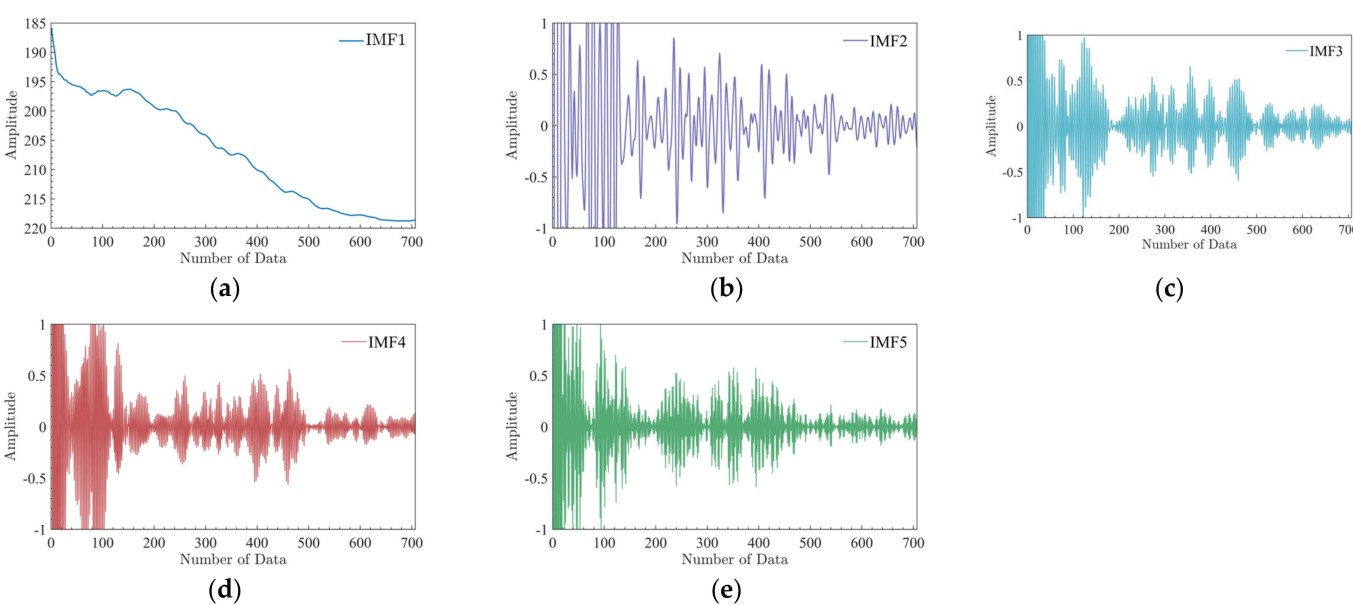

**Figure 8.** IMF components of PL sequence for the first set:(**a**) IMF1; (**b**) IMF2; (**c**) IMF3; (**d**) IMF4; (**e**) IMF5.

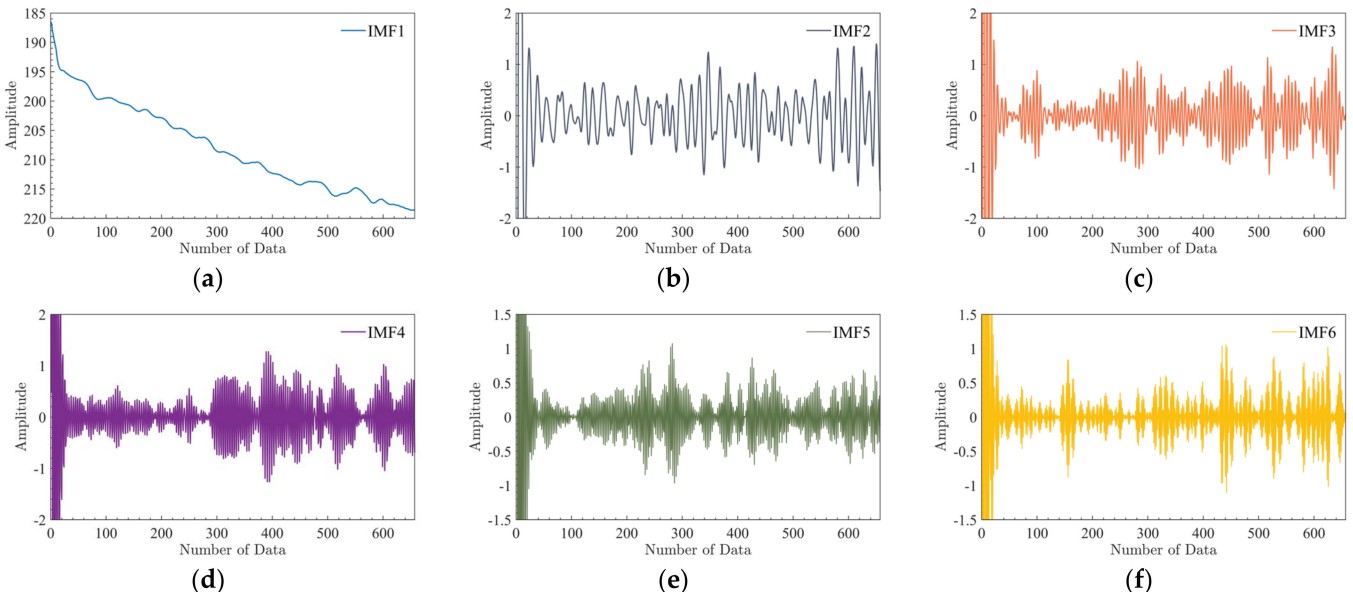

**Figure 9.** IMF components of PL sequence for the second set:(**a**) IMF1; (**b**) IMF2; (**c**) IMF3; (**d**) IMF4; (**e**) IMF5; (**f**) IMF6.

Therefore, the corresponding prediction models are built for multiple IMF components of the decomposition of three sets of measured PL sequences, and finally the corresponding PL prediction results for three sets of PL sequences are reconstructed.

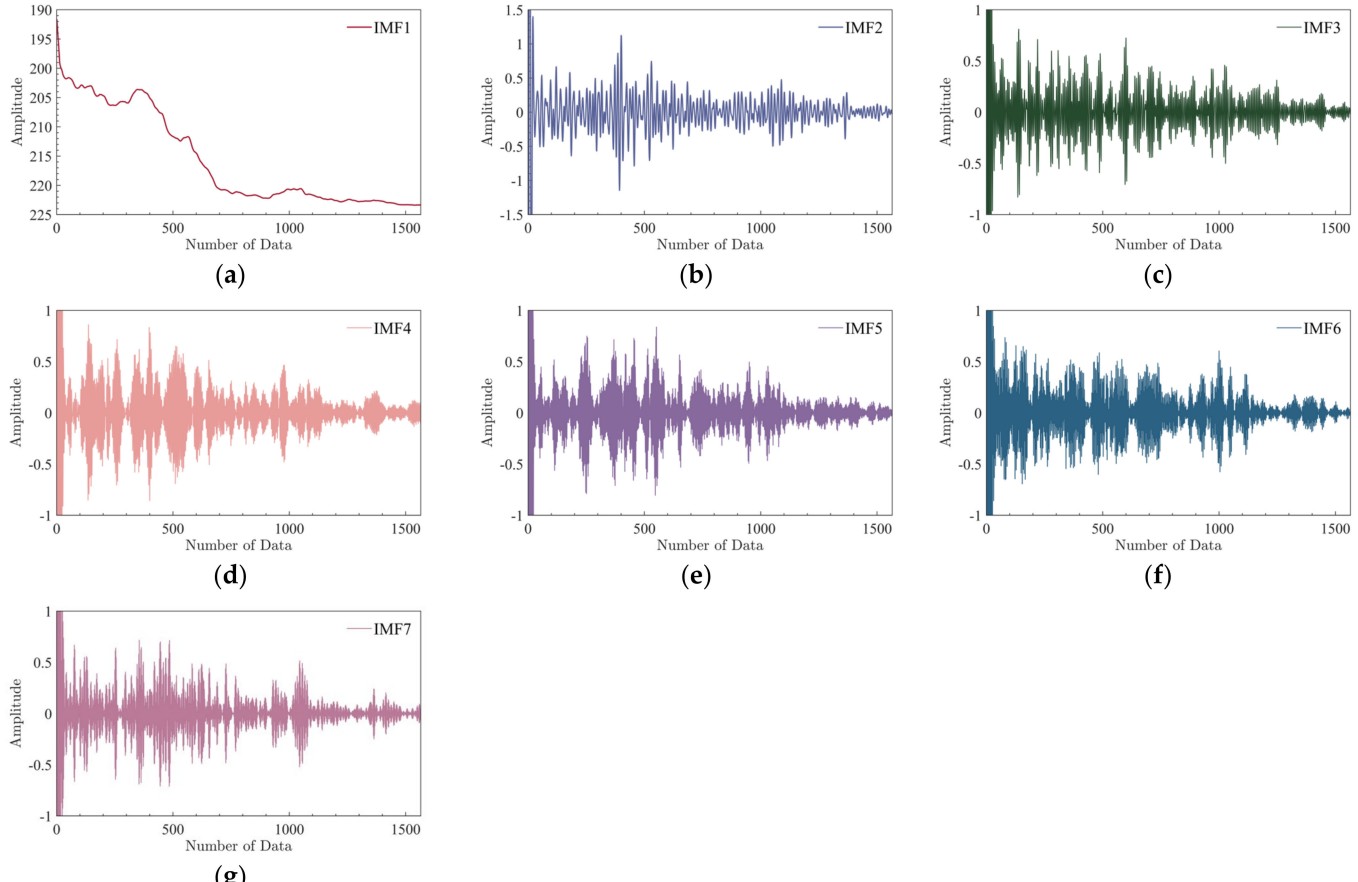

**Figure 10.** IMF components of PL sequence for the third set:(**a**) IMF1; (**b**) IMF2; (**c**) IMF3; (**d**) IMF4; (**e**) IMF5; (**f**) IMF6; (**g**) IMF7.

### 3.3. Building a PL Subsequence Prediction Model Using a LSTM Network

For the VMD−PSO−LSTM method, we use optimized a LSTM network to predict the PL subsequence (IMF component) at long ranges. Since PL values at each measured location have a certain non−linear relationship with both historical and future loss values, it is because of this characteristic that the LSTM network can be used to predict the PL. Additionally, LSTM prediction models are built for each set of PL subsequences. The LSTM prediction model learns the characteristic historical information about the PL subsequence, and the output is the predicted value of the IMF component. Finally, the predicted values of each IMF component are reconstructed to obtain the final PL predicted values [37].

The LSTM network is a special kind of RNN with good generalization ability and fault tolerance, which effectively improves the problems of gradient explosion, gradient disappearance and poor memory that exist in RNN. An LSTM network saves the long−term state by adding a hidden memory unit, the memory unit in LSTM network consists of the input gate, forget gate and output gate [38]. The forget gate is used to select the information to be discarded through the Sigmoid function, the input gate is used to select the current information to be saved and the output gate is used to control the number of information to be output. With the three gate mechanisms, the LSTM network's memory unit can continuously update information at different moments, so the LSTM model is able to learn the long−term evolution regulation of the PL subsequence [39]. The prediction process of the PL subsequence combining LSTM network is shown in Figure 11.

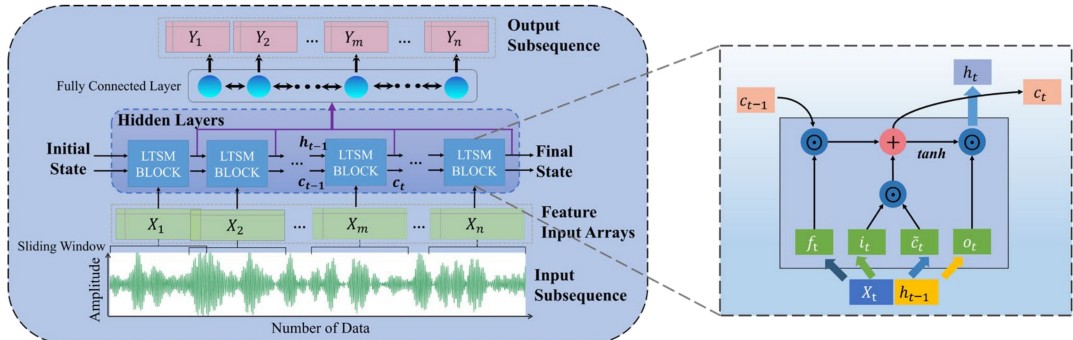

**Figure 11.** Prediction process for the PL subsequence combining LSTM network.

The detailed processes of learning the long−term evolution regulation of the propagation loss subsequence in the memory unit are as follows. The first step of the LSTM network is to determine what information about PL will be discarded from the unit state. This step is determined by the forget gate. The output $h_{t-1}$ at the previous moment and input $X_t$ at the current moment are input into the Sigmoid activation function and output a number between 0 and 1 to each number in the cell state $c_{t-1}$ at the previous moment, 1 represents the "fully keep" state and 0 represents the "fully forget" state. The calculation equation of the forget gate is as follows:

$$f_t = \sigma\left(W_f X_t + U_f h_{t-1} + b_f\right) \tag{13}$$

The next step will determine what PL characteristics information will be saved in the unit state by the input gate. This step consists of two parts. One is that the Sigmoid function determines which values need to be updated, and the other part is that the tanh activation function will generate a new candidate state $\widetilde{c}_t$, which can be added to the memory unit state. The input gate is calculated as follows:

$$i_t = \sigma(W_i X_t + U_i h_{t-1} + b_i) \tag{14}$$

$$\widetilde{c}_t = \tanh(W_c X_t + U_c h_{t-1} + b_c) \tag{15}$$

Then the old cell state $c_{t-1}$ at the previous moment is updated with the new cell state $c_t$. The output of the forget gate is multiplied by the old cell state $c_{t-1}$; the output of the input gate is multiplied by the new candidate information, and the sum of the two parts generates a new cell state $c_t$ as follows:

$$c_t = f_t \cdot c_{t-1} + i_t \cdot \widetilde{c}_t \tag{16}$$

Finally, the output gate determines what information will be output. The output $h_{t-1}$ at the previous moment and input $X_t$ at the current moment are input into the Sigmoid activation function, and the newly obtained cell state $c_t$ is then input into the tanh activation function to set the unit state value between $-1$ and 1; subsequently, the two parts are multiplied as follows:

$$o_t = \sigma(W_o X_t + U_o h_{t-1} + b_o) \tag{17}$$

$$h_t = o_t \cdot \tanh(c_t) \tag{18}$$

where $h_{t-1}$ represents the hidden memory unit state at the previous moment, $X_t$ represents the PL subsequence of the input at current moment, i.e., $IMF(t)$. $h(t)$ represents the output at the current moment. $W_i$, $U_i$, $W_f$, $U_f$, $W_o$, $U_o$, $W_c$ and $U_c$ represent the weight vector matrix corresponding to the three gates; $b_i$, $b_f$, $b_o$ and $b_c$ represent biasing term; $i_t$ represents the input gate, $f_t$ represents the forget gate and $o_t$ represents the output gate.

$\sigma$ and tanh represent the Sigmoid activation function and hyperbolic tangent activation function, respectively, and are calculated as follows:

$$\text{Sigmoid}(X) = \frac{1}{1 + e^{-X}} \tag{19}$$

$$\tanh(X) = \frac{e^X - e^{-X}}{e^X + e^{-X}} \tag{20}$$

In the LSTM network, choosing the appropriate training function in the modeling process has an important influence on the accuracy of the prediction results, and the appropriate training function can make the prediction model provide more ideal results. This study chooses tanh and Sigmoid functions as the activation functions, and Mean Square Error (*MSE*) as the loss function when training the model. This study also uses the Adam method to optimize the loss function in the back propagation of the LSTM network. The equation for the loss function is as follows:

$$Loss = \frac{\sum_{i=1}^{H}\left(y^{pred}(i) - y(i)\right)^2}{H} \tag{21}$$

where $H$ represents the size of the training set, $y^{pred}(i)$ and $y(i)$ represent the predicted result and corresponding true value of the PL, respectively. The modeling process for the PL subsequence prediction model based on an optimized LSTM network is shown in Figure 12.

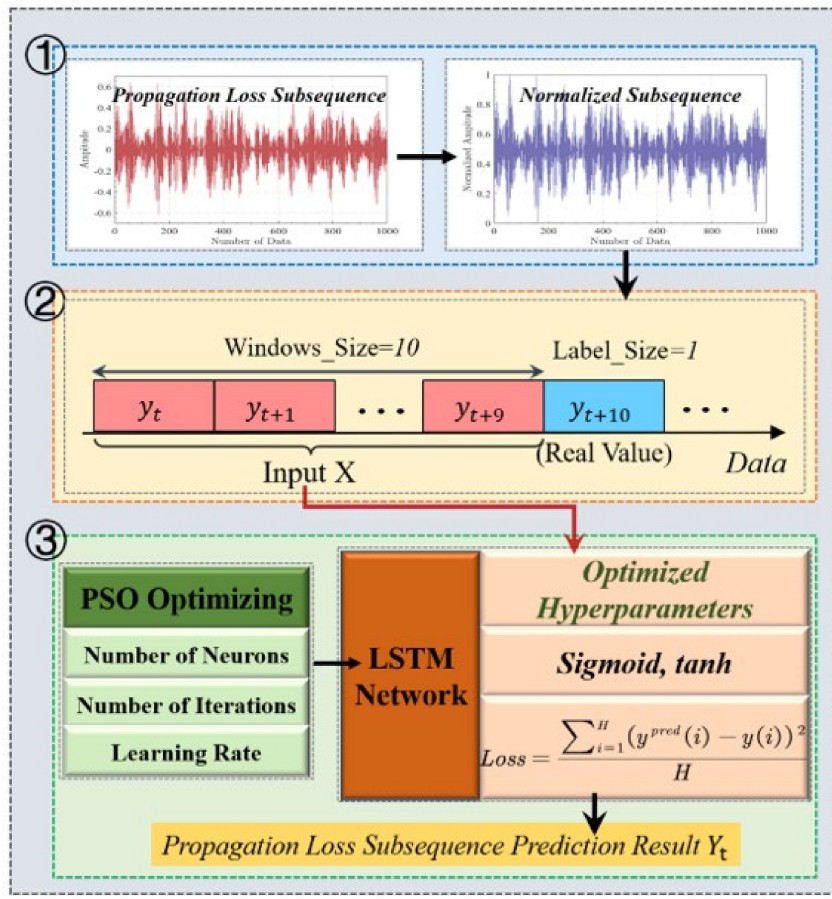

**Figure 12.** Modeling process for PL subsequence prediction model based on optimized LSTM network. (1) Normalizing PL subsequence; (2) setting prediction window to construct training and test sets; (3) using optimized LSTM network for prediction.

For the measured PL, the front 70% of the loss values of each set of subsequences obtained after decomposition are used to construct a training set to train the LSTM network, and the remaining 30% of the loss values are used to test the prediction performance of the VMD−PSO−LSTM method. The size of the prediction window is set to 10 for the PL subsequence prediction, i.e., this study uses the front 10 PL subsequence values to predict the next loss value. Furthermore, to enhance the convergence velocity and prediction accuracy of the LSTM prediction model, we first normalize each PL subsequence before model training, distributing the PL subsequence input to the prediction model between 0 and 1 [40]. The normalized PL subsequence is then predicted by the LSTM network. The normalization is calculated as follows:

$$Y = \frac{X - X_{\min}}{X_{\max} - X_{\min}} \tag{22}$$

where $X_{\max}$ and $X_{\min}$ represent the maximum and minimum values in the PL subsequence, respectively. Furthermore, before training the LSTM network, this study uses the PSO algorithm to optimize the hyperparameters of the LSTM network.

*3.4. Uing PSO Algorithm to Optmize Hyperparameters of LSTM Network*

The LSTM network has obvious advantages in time sequence prediction, but the setting of hyperparameters in a LSTM network is very critical for model training, relying only on empirical selection of parameters will not only enhance the training difficulty of the model, but also result in the trained model not having the best prediction performance. The main purpose of using PSO to optimize the LSTM network is that, before training the LSTM network, instead of artificially setting the hyperparameters of the LSTM network based on experience (the number of neurons, the number of iterations and the learning rate), the three hyperparameters are input as a set into the PSO algorithm for optimization. The PSO algorithm simulates the prediction process of the LSTM network based on the training set, and when the fitness function is minimized, the optimal solution is selected and set to the LSTM network to build a PL subsequence prediction model, so that the final LSTM prediction model has the best prediction performance [41]. The update equations of the PSO algorithm can be expressed as follows:

$$v_{p,m+1} = w \cdot v_{p,m} + c_1 \cdot rand_1 \cdot \left( pbest_p - x_{p,m} \right) \\ + c_2 \cdot rand_2 \cdot \left( gbest_m - x_{p,m} \right) \tag{23}$$

$$x_{p,m+1} = x_{p,m} + v_{p,m+1} \tag{24}$$

where $w$ represents the inertia weight. $c_1$ and $c_2$ represent learning factors for particle and population, respectively. $rand_1$ and $rand_2$ represent two random values between 0 and 1. $v_{p,m}$ and $x_{p,m}$ represent the velocity and position of the particle $p$ at the $m$th iteration, respectively. $pbest_p$ and $gbest_m$ represent the known optimal solutions for particle and population, respectively.

The core idea of the PSO algorithm is to consider each hyperparameter of LSTM network to be optimized as a particle in the search space. Each particle has two properties: velocity and position. Position is used to describe the current state of the particle and velocity is used to describe the movement of the particle at the next iteration. The particle finds the optimal solution by updating its own velocity and position according to Equations (23) and (24). In each iteration, the particle updates itself by tracking two extreme values: the first is the optimal solution currently found by the particle, called individual optimal position, and the other is the optimal solution currently found in the whole population, called population optimal position [42]. The search process for the PSO algorithm in this study is shown in Figure 13.

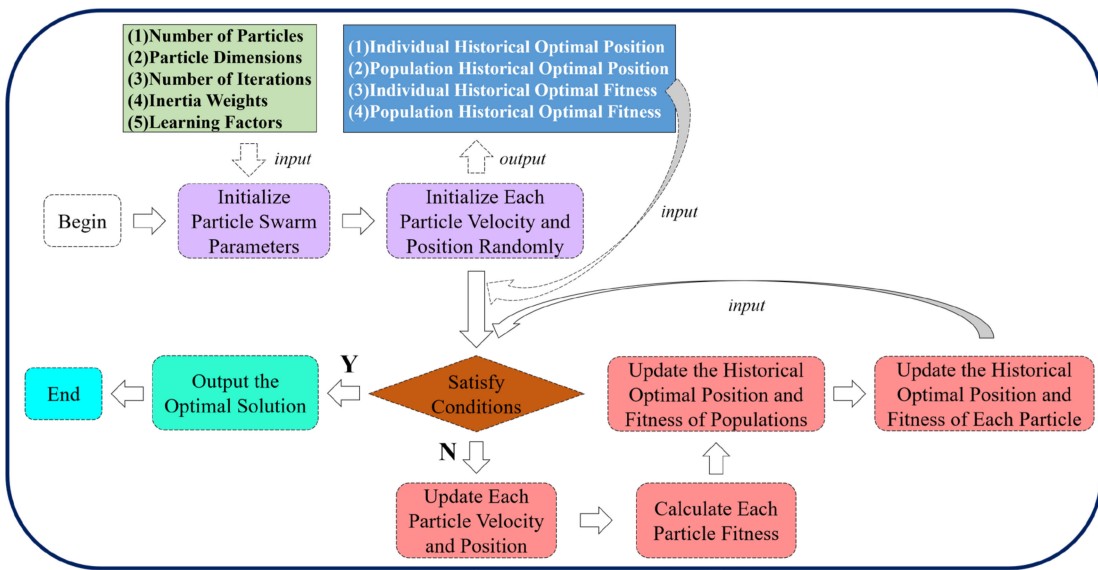

**Figure 13.** Search process for the PSO algorithm.

Based on PL subsequences, the velocity and position of each particle need to be randomly initialized. The particle swarm related parameters need to be initialized before the hyperparameters of LSTM network are optimized. In this study, the parameters of the PSO algorithm are set as follows: the number of evolutionary iterations is 100, the number of particles is 30, the particle dimension is 3, the learning factors $c_1 = c_2 = 1.49$, the inertia weight $w = 0.8$ and the *MSE* function is used as the fitness function.

According to the particle dimension and the hyperparameters of the LSTM network to be optimized, the expression for the initialized particle is as follows:

$$x_{i,0} = (h, \varepsilon, n_{LSTM}) \tag{25}$$

where $h$ represents the number of neurons, $\varepsilon$ represents the learning rate and $n_{LSTM}$ represents the number of iterations. Based on certain experience of setting hyperparameters, the three hyperparameters $h$, $\varepsilon$ and $n_{LSTM}$ in $x_{i,0}$ set in this study are set in the range [1, 300], [1, 500] and [0.001, 0.01], respectively, for the optimal search [43]. In this study, based on the LSTM prediction model built from the PL subsequences, the hyperparameters of the optimal LSTM prediction model optimized by the PSO algorithm are shown in Tables 7–9.

**Table 7.** Results of hyperparameters optimized by PSO based on first set PL.

| PL Subsequence First Set | Number of Neurons | Number of Iterations | Learning Rate |
|---|---|---|---|
| IMF1 | 90 | 228 | 0.0091 |
| IMF2 | 65 | 290 | 0.0067 |
| IMF3 | 157 | 383 | 0.0078 |
| IMF4 | 210 | 169 | 0.0072 |
| IMF5 | 212 | 203 | 0.0094 |

**Table 8.** Results of hyperparameters optimized by PSO based on second set PL.

| PL Subsequence Second Set | Number of Neurons | Number of Iterations | Learning Rate |
|---|---|---|---|
| IMF1 | 219 | 243 | 0.0049 |
| IMF2 | 92 | 470 | 0.0084 |
| IMF3 | 79 | 420 | 0.0043 |
| IMF4 | 193 | 376 | 0.0074 |
| IMF5 | 144 | 151 | 0.0091 |
| IMF6 | 29 | 309 | 0.0074 |

**Table 9.** Results of hyperparameters optimized by PSO based on third set PL.

| PL Subsequence Third Set | Number of Neurons | Number of Iterations | Learning Rate |
|---|---|---|---|
| IMF1 | 281 | 254 | 0.0070 |
| IMF2 | 82 | 134 | 0.0074 |
| IMF3 | 25 | 433 | 0.0086 |
| IMF4 | 137 | 178 | 0.0060 |
| IMF5 | 262 | 373 | 0.0082 |
| IMF6 | 117 | 193 | 0.0081 |
| IMF7 | 39 | 443 | 0.0075 |

## 4. Prediction Results and Analysis

### 4.1. Performance Evaluation Indicators

To evaluate the prediction performance of the VMD−PSO−LSTM method, this study uses Root Mean Square Error (*RMSE*) and Mean Absolute Error (*MAE*) to evaluate the prediction performance of the VMD−PSO−LSTM method from different perspectives based on the test set. The smaller the value of *RMSE* and *MAE*, the better the prediction performance of the VMD−PSO−LSTM method and the higher the prediction accuracy of the PL. The evaluation indicators *RMSE* and *MAE* are calculated as follows:

$$MAE = \frac{\sum_{i=1}^{T} \left| L^{pred}(i) - L(i) \right|}{T} \tag{26}$$

$$RMSE = \sqrt{\frac{\sum_{i=1}^{T} \left( L^{pred}(i) - L(i) \right)^2}{T}} \tag{27}$$

where $L^{pred}(i)$ represents the predicted result of the PL sequence, $L(i)$ represents the true value of the PL and $T$ represents the size of the test set.

In addition, to compare the prediction performance of the VMD−PSO−LSTM method with other prediction methods, this study uses $P_{RMSE}$ and $P_{MAE}$ to represent the prediction improved performance of the VMD−PSO−LSTM method in *RMSE* and *MAE*, respectively [33], $P_{RMSE}$ and $P_{MAE}$ are calculated as follows:

$$P_{RMSE} = \frac{|RMSE_1 - RMSE_2|}{RMSE_1} \times 100\% \tag{28}$$

$$P_{MAE} = \frac{|MAE_1 - MAE_2|}{MAE_1} \times 100\% \tag{29}$$

### 4.2. PL Prediction Results and Analysis

To show the advantages of the VMD−PSO−LSTM method in PL prediction, this study additionally builds another seven prediction models for comparison with the VMD−PSO−LSTM method to verify its performance. These include single prediction methods: the RNN network, gate recurrent unit (GRU) network and LSTM network. These also include hybrid prediction methods: the PSO−LSTM method, VMD−LSTM method, a method (WST−LSTM) consisting of wavelet scattering transform (WST) and LSTM and a method (VMD−GA−LSTM) consisting of a genetic algorithm (GA), VMD and LSTM [44]. Three sets of measured PL sequences are input into the above seven prediction models and the VMD−PSO−LSTM model for training and prediction. Their prediction performance is evaluated using *RMSE* and *MAE* indicators. The Figures 14–16 show the prediction results of the multiple prediction methods on three sets of PL test sets. It can be seen from Figures 14–16 that as the propagation range varies, the PL sequences predicted by the RNN, GRU and LSTM networks are farther away from the PL and are greater than the PL of the first and third sets. In addition, the RNN, GRU and LSTM networks have worse prediction results when the PL sequence has large fluctuations. The predicted sequences

of the other methods (WST−LSTM, VMD−LSTM, PSO−LSTM, VMD−GA−LSTM and VMD−PSO−LSTM methods) fluctuate around the PL. For the first set of PL, the prediction results of WST−LSTM, VMD−LSTM and PSO−LSTM methods also deviate from PL below the PL sequence.

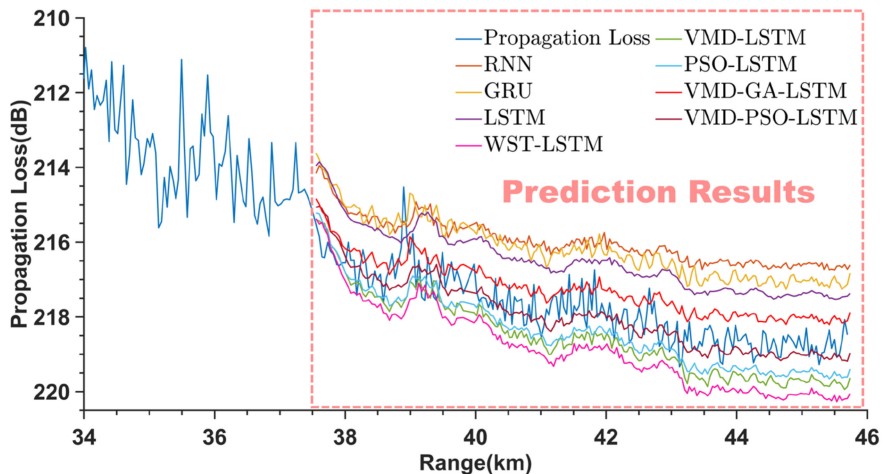

**Figure 14.** Prediction results of the first set of PL for different methods.

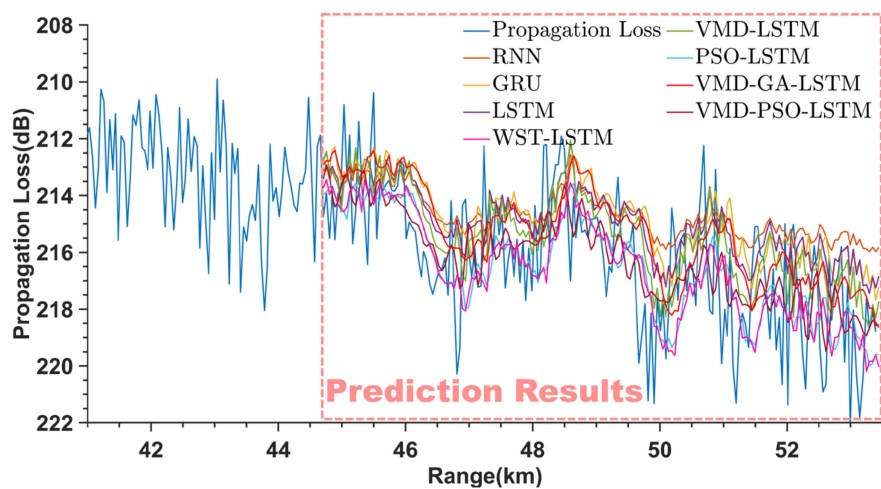

**Figure 15.** Prediction results of the second set of PL for different methods.

The WST−LSTM, VMD−LSTM and PSO−LSTM methods can only weakly reflect the trend of the PL sequence, compared with the prediction results of RNN, GRU and LSTM networks. The VMD−GA−LSTM method cannot fully reflect the direction of the trend and fluctuation size of the PL sequence correctly. However, the trend situation of its prediction results is generally consistent with the real situation. The VMD−PSO−LSTM has the best prediction results, and the PL sequence predicted by the VMD−PSO−LSTM method at more remote locations has better fitting results compared with other methods, which can better reflect the propagation trend and fluctuation size of the PL.

Based on the three sets of PL sequences, the prediction results of the VMD−PSO−LSTM method and other seven prediction methods are compared. This study calculates the *RMSE* and *MAE* indicators of the VMD−PSO−LSTM method and the other seven prediction methods in the test set to evaluate the prediction performance from different perspectives. The *RMSE* and *MAE* indicators of different prediction methods in each PL test set are shown in Tables 10 and 11.

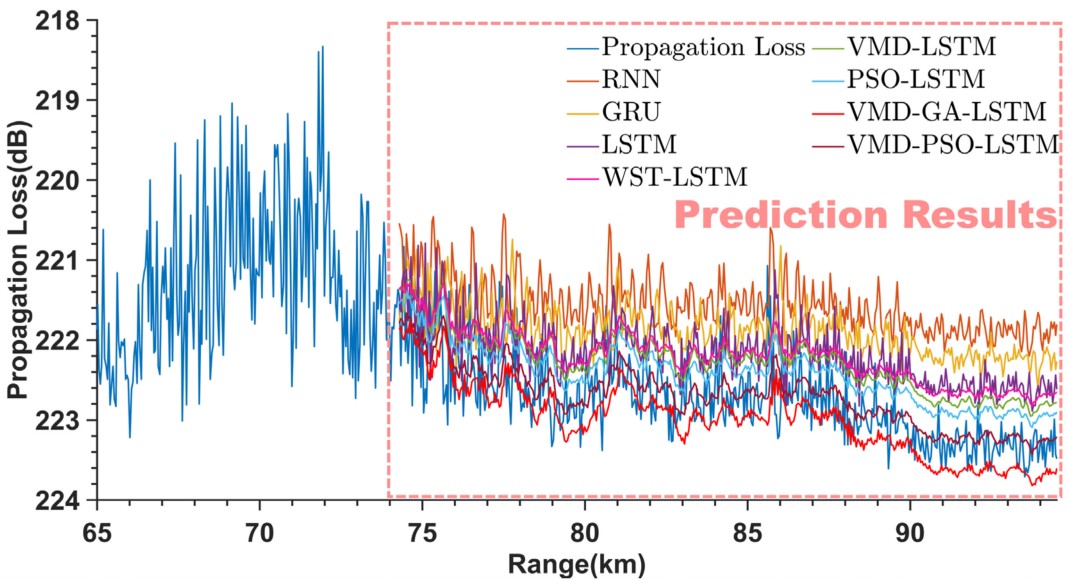

**Figure 16.** Prediction results of the third set of PL for different methods.

**Table 10.** *RMSE* of different prediction methods for three sets of PL sequences.

| Prediction Method | RMSE (dB) | | |
|---|---|---|---|
| | First Set | Second Set | Third Set |
| RNN | 1.877 | 2.536 | 1.243 |
| GRU | 1.662 | 2.414 | 0.974 |
| LSTM | 1.364 | 2.259 | 0.742 |
| WST−LSTM | 1.213 | 2.165 | 0.667 |
| VMD−LSTM | 0.932 | 2.010 | 0.609 |
| PSO−LSTM | 0.783 | 1.909 | 0.497 |
| VMD−GA−LSTM | 0.704 | 1.878 | 0.447 |
| VMD−PSO−LSTM | 0.517 | 1.682 | 0.368 |

**Table 11.** *MAE* of different prediction methods for three sets of PL sequences.

| Prediction Method | MAE (dB) | | |
|---|---|---|---|
| | First Set | Second Set | Third Set |
| RNN | 1.813 | 2.051 | 1.175 |
| GRU | 1.606 | 1.951 | 0.892 |
| LSTM | 1.292 | 1.836 | 0.659 |
| WST−LSTM | 1.105 | 1.762 | 0.604 |
| VMD−LSTM | 0.816 | 1.614 | 0.544 |
| PSO−LSTM | 0.665 | 1.517 | 0.428 |
| VMD−GA−LSTM | 0.613 | 1.476 | 0.351 |
| VMD−PSO−LSTM | 0.406 | 1.332 | 0.276 |

As Tables 10 and 11 show, both *RMSE* and *MAE* indicators of the VMD−PSO−LSTM method are smaller than the other seven comparison methods for different PL, which shows that the prediction performance of the VMD−PSO−LSTM method is better than that of the other comparison methods, the VMD−PSO−LSTM method has a higher PL prediction accuracy than the other comparison methods.

Tables 12 and 13 show the percentage improvement in the PL prediction performance of the VMD−PSO−LSTM method compared to other seven methods. The advantage of the VMD−PSO−LSTM method is further verified by the $P_{RMSE}$ and $P_{MAE}$ indicators. According to the comparison results, the VMD−PSO−LSTM method has lower error and higher

prediction accuracy for PL prediction. Compared to the other seven prediction methods, for three sets of measured PL sequences, the VMD−PSO−LSTM method improves the prediction performance at most by 72.46 and 77.61%, and improves by at least 10.44 and 9.76% in *RMSE* and *MAE*, respectively. Moreover, the VMD−PSO−LSTM model has good generalization which has better prediction performance in different sets of PL sequences.

**Table 12.** $P_{RMSE}$ of the VMD−PSO−LSTM method in comparison with the other seven methods.

| Prediction Method | $P_{RMSE}$ | | |
|:---:|:---:|:---:|:---:|
| | First Set | Second Set | Third Set |
| RNN | 72.46% | 33.68% | 70.39% |
| GRU | 68.89% | 30.32% | 62.22% |
| LSTM | 62.10% | 25.54% | 50.40% |
| WST−LSTM | 57.38% | 22.31% | 44.83% |
| VMD−LSTM | 44.53% | 16.32% | 39.57% |
| PSO−LSTM | 33.97% | 11.89% | 25.96% |
| VMD−GA−LSTM | 26.56% | 10.44% | 17.67% |

**Table 13.** $P_{MAE}$ of the VMD−PSO−LSTM method in comparison with the other seven methods.

| Prediction Method | $P_{MAE}$ | | |
|:---:|:---:|:---:|:---:|
| | First Set | Second Set | Third Set |
| RNN | 77.61% | 35.06% | 76.51% |
| GRU | 74.72% | 31.73% | 69.06% |
| LSTM | 68.58% | 27.45% | 58.12% |
| WST−LSTM | 63.26% | 24.40% | 54.30% |
| VMD−LSTM | 50.25% | 17.47% | 49.26% |
| PSO−LSTM | 38.95% | 12.20% | 35.51% |
| VMD−GA−LSTM | 33.77% | 9.76% | 21.37% |

To show prominently the influences of the VMD method, the PSO algorithm was used on the PL prediction results. The performance of the VMD−PSO−LSTM method still requires further analysis from the prediction results. This study compares the *RMSE* and *MAE* indicators of six prediction methods (LSTM, WST−LSTM, VMD−LSTM, PSO−LSTM, VMD−GA−LSTM and VMD−PSO−LSTM), and the two indicators for three sets of PL sequences are shown in Figure 17. The following correlation analysis is conducted:

**(1)** To analyze the influence of the introducing of the VMD on the PL prediction performance, this study compares the prediction performance of VMD−LSTM and LSTM, the prediction performance of VMD−PSO−LSTM and PSO−LSTM and the prediction performance of VMD−LSTM and WST−LSTM. For the three sets of PL sequences, the VMD−LSTM improves the prediction performance compared with the LSTM at most by 31.67 and 36.84%, and improves by at least 11.02 and 12.09% in *RMSE* and *MAE*, respectively. The VMD−PSO−LSTM improves the prediction performance compared with the PSO−LSTM at most by 33.97 and 38.95%, and improves by at least 11.89 and 12.20% in *RMSE* and *MAE*, respectively. Finally, the VMD−LSTM improves the prediction performance compared with the WST−LSTM at most by 23.17 and 26.15%, and improves by at least 7.16 and 8.40% in *RMSE* and *MAE*, respectively. The above analysis results show that the introduction of the VMD effectively lowers the non−smoothness and complexity of PL, reduces the prediction errors of the PL subsequences and effectively improves the prediction performance and practicality of the LSTM. Furthermore, the VMD method is not only effective in improving the prediction accuracy of the single LSTM prediction model, but also still helps significantly in improving the prediction accuracy of the hybrid model.

**(2)** To analyze the influence of the introducing of the PSO on the PL prediction performance, this study compares the prediction performance of PSO−LSTM and LSTM, the

prediction performance of VMD−PSO−LSTM and VMD−LSTM and the prediction performance of VMD−PSO−LSTM and VMD−GA−LSTM. For the three sets of PL sequences, the PSO−LSTM improves the prediction performance compared with the LSTM at most by 42.60 and 48.53%, and improves by at least 15.49 and 17.37% in *RMSE* and *MAE*, respectively. The VMD−PSO−LSTM improves the prediction performance compared with the VMD−LSTM at most by 44.53 and 50.25%, and improves by at least 16.32 and 17.47% in *RMSE* and *MAE*, respectively. Finally, the VMD−PSO−LSTM improves the prediction performance compared with the VVMD−GA−LSTM at most by 26.56 and 33.77%, and improves by at least 10.44 and 9.76% in *RMSE* and *MAE*, respectively. The above analysis results show that the introduction of the PSO has greater effectiveness in improving the prediction performance of the single LSTM network and the hybrid VMD−LSTM method. Using the hyperparameters of the LSTM optimized by the PSO, the mapping relationship between the historical and future information of PL can be better built so that the LSTM can converge better. Thus the prediction model of the LSTM can be better built.

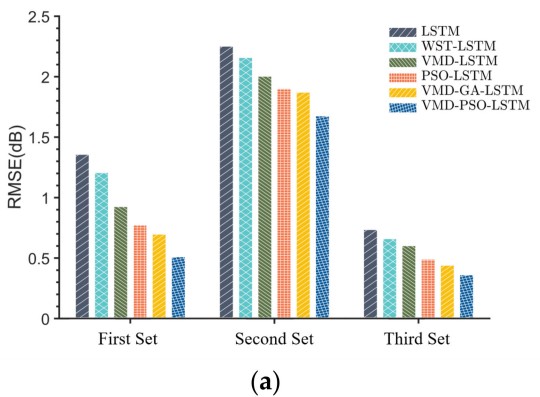
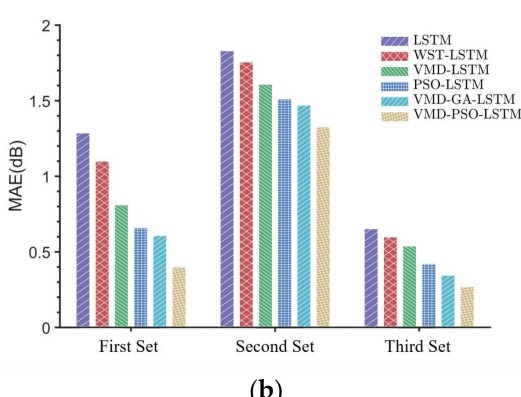

|   |   |
|---|---|
| (a) | (b) |

**Figure 17.** *RMSE* and *MAE* indicators of the six methods (LSTM, WST−LSTM, VMD−LSTM, PSO−LSTM, VMD−GA−LSTM and VMD−PSO−LSTM methods): (**a**) *RMSE*; (**b**) *MAE*.

## 5. Conclusions and Future Work

To achieve accurate prediction of EM waves' PL in EDs, a multiscale decomposition prediction model (VMD−PSO−LSTM) for PL prediction is proposed. Firstly, the VMD method is used to decompose the measured PL, which can effectively solve the problems such as high randomness and non−linearity of the PL. Through the LSTM network, the non−linear mapping relationships between the historical and future information about PL are learned. Additionally, the PSO algorithm is introduced to optimize the hyper-parameters of the LSTM network, so that the model can converge better and lower the training complexity. To verify the prediction performance of the proposed method, the VMD−PSO−LSTM method is compared with the other seven methods. By comparing the prediction results of three sets of PL sequences, the following conclusions are obtained based on the prediction results: (1) the VMD−PSO−LSTM method performs significantly better than other seven methods in *RMSE* and *MAE* indicators, indicating that the pro-posed method can satisfy the requirements of accurate prediction application for PL; (2) the VMD method can enhance the PL prediction performance and accuracy of the above non−decomposition methods (LSTM and PSO−LSTM); (3) the PSO algorithm can enable LSTM to achieve better convergence and further enhance the prediction performance of the LSTM model significantly.

In future work, we will focus more on building a more effective hybrid prediction model combining VMD−PSO−LSTM built in this study to make a higher accuracy for PL prediction. Additionally, we will consider more spatial information on PL.

**Author Contributions:** Conceptualization, H.J. and J.Z.; methodology, B.Y.; software, H.J.; validation, H.J., B.Y. and J.Z.; formal analysis, Y.Z.; investigation, Q.L., C.H.; writing—original draft preparation, H.J., B.Y. and J.Z.; writing—review and editing, Y.Z. and Q.L.; project administration, Y.Z.; funding acquisition, Q.L. All authors have read and agreed to the published version of the manuscript.

**Funding:** This research was funded by the National Natural Science Foundation of China, grant number 62271457 and the Key R&D Project of Shandong Province, grant numbers 2019JMRH0109 and 2020JMRH0201.

**Institutional Review Board Statement:** Not applicable.

**Informed Consent Statement:** Not applicable.

**Data Availability Statement:** Not applicable.

**Acknowledgments:** Authors thank all the editors and reviewers for their valuable comments that greatly improved the presentation of this paper. Authors are very grateful to China Research Institute of Radiowave Propagation for providing the measured propagation loss.

**Conflicts of Interest:** The authors declare no conflict of interest.

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
