# Peer review of "Multiscale Decomposition Prediction of Propagation Loss for EM Waves in Marine Evaporation Duct Using Deep Learning"

_jmse, doi:10.3390/jmse11010051_

Round 1
Reviewer 1 Report
See applied file

Reviewer 2 Report
The paper proposes an interesting hybrid method for propagation loss prediction by combining VMD method, LSTM network and PSO algorithm based on measured propagation loss for accurate prediction in marine evaporation duct. The paper is technically sound and well writen. It would be more convincing to consider the following revision:
1) It would be interesting to see the comparion results for the hybrid method consists of wavelet scattering transform (WST) followed by LSTM network.
Reviewer 3 Report
The authors need to look at the following paper:
Dang, M.; Wu, J.; Cui, S.; Guo, X.; Cao, Y.; Wei, H.; Wu, Z. Multiscale Decomposition Prediction of Propagation Loss in Oceanic Tropospheric Ducts. Remote Sens. 2021, 13, 1173. https://doi.org/10.3390/rs13061173
The research presented in this paper is very similar. It isn't easy to judge the quality of the VMD-PSO-LSTM method presented here compared to other previous models, including the citation above. I recommend the authors show the novelty in their approach more clearly. For example, if they chose VMD-GA-LSTM, would that give a better result? I suggest GA because it was investigated earlier (citation above). Was there a reason to select PSO? Right now, the authors only describe the improvements from their choices (Figure 17, for example).
Reviewer 4 Report
The authors propose a new deep learning method for prediction of propagation loss of EM waves with evaporation duct loss as the experimental example.
The new prediction model is named as VMD-PSO-LSTM which is a combination variational mode decomposition method (VMD), particle swarm optimization (PSO) algorithm and long short-term memory (LSTM) network. To show its advantage the new method is compared with other common prediction models and truly lower error is achieved.
Overall while the content of the manuscript is great and can be quite useful for the scientific community of this field the writing style should be improved and made more compact. For example while spacings are used some sentences are just too long. The expression "in this study" is repeated 27 times. Paragraph from line 334 to 346 contains multiple rules which could be shown differently.
Small comments/corrections:
1. Please mention what does IMF acronym stand for.
2. Fig.3b is shown in page 4 but referred in page 6 (typically referred on same or next page).
3. If possible please describe more clearly what I should look in Fig.4 & Fig.5.
4. What is the run times for the considered method and other methods?
5. In Fig.14 to Fig.16 the new method loss is quite smooth. Could this be also the actual realistic result? As in do we not lose any actual information such a way?
Round 2
Reviewer 2 Report
The paper has been improved and revised accordingly.
Author Response
Dear reviewer,
Thank you very much for your comments and professional advice. These opinions help to improve academic rigor of our manuscript. Based on your suggestions and opinions, we have made corrected modifications on the revised manuscript. Terms that are repeated several times have been abbreviated or removed. Long sentences have been divided into more short sentences. We have carefully spell-checked and corrected the language in the manuscript. Now the article is presented with easy for reading text.
Reviewer 3 Report
I'm satisfied with the author response.
Author Response

(The authors gave the same response as above.)

Reviewer 4 Report
I can agree that the writing has been improved compared to the previous version.
*In the obtained manuscript version a small mistake:
Chapter 2 title was at the end of page 3 and table 1 at the end of page 4. Please shift both to the next page.
Author Response
Dear reviewer,
Thank you very much for your comments and professional advice. These opinions help to improve academic rigor of our manuscript. Based on your suggestions and opinions, we have made corrected modifications on the revised manuscript. We have carefully spell-checked and corrected the language in the manuscript. The chapter 2 title and Table 1 have both been moved to the next page. Now the article is presented with easy for reading text.